# A Cross-Stitch Architecture for Joint Registration and Segmentation in Adaptive Radiotherapy

**Laurens Beljaards**[1]                                   L.R.BELJAARDS@UMAIL.LEIDENUNIV.NL

**Mohamed S. Elmahdy**[2]                                      M.S.E.ELMAHDY@LUMC.NL

**Fons Verbeek**[1]                                   F.J.VERBEEK@LIACS.LEIDENUNIV.NL

**Marius Staring**[2,3]                                        M.STARING@LUMC.NL

[1] *Leiden Institute of Advanced Computer Science, 2333 CA Leiden, The Netherlands*

[2] *Division of Image Processing, Department of Radiology, Leiden University Medical Center, 2300 RC Leiden, The Netherlands*

[3] *Department of Radiation Oncology, Leiden University Medical Center, 2300 RC Leiden, The Netherlands*

## Abstract

Recently, joint registration and segmentation has been formulated in a deep learning setting, by the definition of joint loss functions. In this work, we investigate joining these tasks at the architectural level. We propose a registration network that integrates segmentation propagation between images, and a segmentation network to predict the segmentation directly. These networks are connected into a single joint architecture via so-called cross-stitch units, allowing information to be exchanged between the tasks in a learnable manner. The proposed method is evaluated in the context of adaptive image-guided radiotherapy, using daily prostate CT imaging. Two datasets from different institutes and manufacturers were involved in the study. The first dataset was used for training (12 patients) and validation (6 patients), while the second dataset was used as an independent test set (14 patients). In terms of mean surface distance, our approach achieved $1.06 \pm 0.3$ mm, $0.91 \pm 0.4$ mm, $1.27 \pm 0.4$ mm, and $1.76 \pm 0.8$ mm on the validation set and $1.82 \pm 2.4$ mm, $2.45 \pm 2.4$ mm, $2.45 \pm 5.0$ mm, and $2.57 \pm 2.3$ mm on the test set for the prostate, bladder, seminal vesicles, and rectum, respectively. The proposed multi-task network outperformed single-task networks, as well as a network only joined through the loss function, thus demonstrating the capability to leverage the individual strengths of the segmentation and registration tasks. The obtained performance as well as the inference speed make this a promising candidate for daily re-contouring in adaptive radiotherapy, potentially reducing treatment-related side effects and improving quality-of-life after treatment.

**Keywords:** Multi-Organ Segmentation, Deformable Registration, Adaptive Radiotherapy, Contour Propagation, Convolutional Neural Networks (CNN), Multi-Task Learning (MTL).

## 1. Introduction

Adaptive image-guided radiation therapy aims to adapt the radiation dose to the daily anatomy of the patient. When executed properly, this may allow the use of smaller safety margins in radiotherapy and thus reduce the exposure of surrounding healthy tissue to

radiation, thereby potentially reducing treatment-related side effects and improving quality of life after treatment. To enable such an adaptive treatment cycle it is required to re-image the patient on a daily basis, and subsequently re-contour the tumor and organs-at-risk. Since contouring takes a substantial amount of time by highly qualified radiation oncologists, adaptive treatment by manual procedures is generally infeasible. Thus, automating the procedure is crucial. The two prevalent methods for automatically contouring medical images are image segmentation and contour propagation using registration. In the context of adaptive image-guided radiotherapy, registration-based methods have the advantage of using prior knowledge of the patient's anatomy in the form of the manually delineated planning scan, and being able to accurately deform low-contrast structures that are hard to identify using nearby higher-contrast structures. Image segmentation has advantages of its own, most notably the ability to accurately contour organs that drastically vary in shape between visits, such as the bladder.

In an attempt to exploit the unique advantages of both methods, approaches for joint registration and segmentation (JRS) have been proposed. Earlier methods performed joint registration and segmentation using for example active contours (Yezzi et al., 2003) or Bayesian models (Pohl et al., 2006). More recently, convolutional neural networks have become prevalent in medical imaging due to the rapid advancements in machine learning. Registration networks can now match the accuracy of iterative approaches, and segmentation networks have already been found to perform better than their conventional counterparts (Litjens et al., 2017). Several approaches have been proposed for joint registration and segmentation in combination with convolutional neural networks. In (Xu and Niethammer, 2019), a framework was presented for jointly training registration and segmentation networks. Other approaches have employed generative adversarial networks for joint registration and segmentation (Mahapatra et al., 2018; Elmahdy et al., 2019b).

In this work we propose to join registration and segmentation further by merging the two tasks at the architectural level rather than only through the loss function, using concepts from the field of multi-task learning (Ruder, 2017). In our novel approach, a single neural network is trained to both propagate contours through image registration and generate contours through image segmentation at the same time. We demonstrate that joint architectures outperform single-task segmentation and registration networks, and we show that our approach generates more accurate organ delineations than state-of-the-art methods on both our validation set and an independent test set of prostate CT scans in terms of median MSD.

## 2. Methods

### 2.1. Base Network Architecture

We use a 3D deep convolutional neural network derived from the U-Net (Ronneberger et al., 2015) and inspired by (Fan et al., 2018) as a base architecture for all networks presented in this paper. The network uses $3 \times 3 \times 3$ convolution layers without padding. LeakyReLU and batch normalization are applied after each convolution layer. Strided convolutions are used for downsampling in the contracting path, and upsampling layers are used for the expansive path. The network has three output resolutions and is deeply supervised at each resolution. At the lower resolution the network can focus on large organs or large deformations and vice

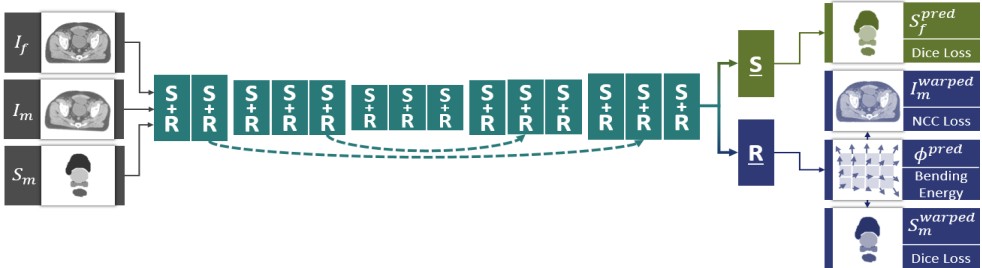

Figure 1: The inputs, architecture, outputs, and the losses of the fully hard parameter sharing network. Here, S stands for the segmentation layer, R for the registration layer, and S+R for shared layer. Only the highest resolution $1 \times 1 \times 1$ convolution layers and outputs are shown here for the sake of clarity.

versa at higher resolutions. Given an input patch of size $n^3$, the sizes of the high, middle, and low resolution patches are $(n-40)^3$, $(\frac{n}{2}-18)^3$, and $(\frac{n}{4}-7)^3$, respectively, where $n = 96$ is used in this paper. A detailed schematic is given in the appendix.

## 2.2. Single-Task Segmentation and Registration Networks

Single-task segmentation and registration networks were trained to serve as a baseline for the performance of the proposed joint networks. These networks have identical architectures except for the input layers and output layers. The segmentation network takes the daily CT scan as input, which we refer to as the fixed image $I_f$, and predicts the corresponding segmentation $S_f^{pred}$. The segmentation network is trained using the Dice Similarity Coefficient (DSC) loss, which quantifies the overlap between $S_f^{pred}$ and the ground truth segmentation $S_f$. The registration network takes both the planning scan, which we refer to as the moving image $I_m$, and the daily scan $I_f$ as input and establishes the correspondence between the two images in the form of a Deformation Vector Field (DVF, $\phi^{pred}$). For this purpose, it is crucial that corresponding anatomical features in the two scans fit inside the network's field of view, therefore the images have been affinely aligned beforehand. The predicted DVF $\phi^{pred}$ is then used to warp $I_m$ such that ideally, the warped moving image $I_m^{warped}$ is identical to $I_f$. The registration network is trained using the Normalized Cross-Correlation (NCC) loss that quantifies the dissimilarity between $I_m^{warped}$ and $I_f$, and the bending energy loss as a regularization term to encourage smoothness of $\phi^{pred}$.

## 2.3. Joining Registration and Segmentation via the Loss

Similar to previous work (Elmahdy et al., 2019b), the network in this approach joins registration and segmentation through the loss function. The network is relatively similar to the registration network discussed in the previous section, with the addition that it also takes $S_m$ as input and is jointly trained using a segmentation Dice loss in addition to the NCC and bending energy losses. This Dice loss penalizes discrepancies between the fixed ground truth segmentation $S_f$ and the warped moving segmentation $S_m^{warped}$.

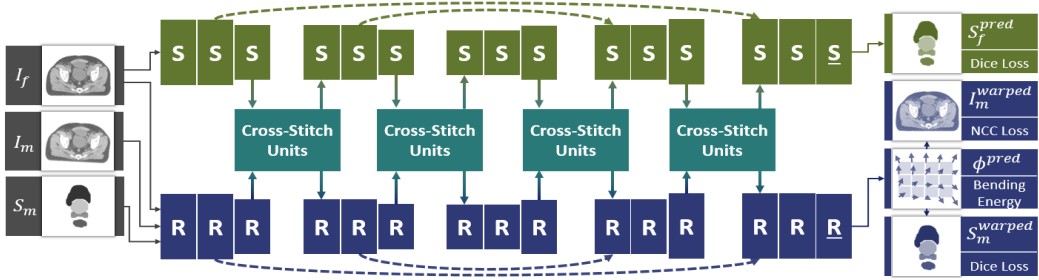

Figure 2: The inputs, architecture, outputs, and losses of the cross-stitch network.

### 2.4. Joint Registration and Segmentation using Hard Parameter Sharing

In this joint network, see Figure 1, the registration and segmentation sub-networks share all their parameters, except for the task-specific $1 \times 1 \times 1$ convolution layers. Apart from these two layers, the network is architecturally similar to the single-task networks. The network is trained with the Dice loss on the segmentation output (similar to the segmentation network), and the NCC, bending energy, and Dice losses on the registration output (similar to the JRS-registration network). Since the network predicts two segmentation maps, one for each path, the contours from one path can be discarded. A simple strategy is to keep the contours from the path that performed best on the validation set. The segmentations can also be selected on a per-organ basis.

### 2.5. Joint Registration and Segmentation via a Cross-Stitch Network

We propose to architecturally join 3D Unet-like networks for registration and segmentation by connecting the paths using cross-stitch units (Misra et al., 2016). The cross-stitch units linearly combine pairs of feature maps from the segmentation path and the registration path using learnable parameters $\boldsymbol{\alpha}$. Given the segmentation path $S$ and the registration path $R$ of the joint network, the feature maps of filter $k$ in layer $\ell \in \{3, 6, 9, 12\}$ – named $X_S^{\ell,k}$ and $X_R^{\ell,k}$ respectively – are connected to a cross-stitch unit with learnable parameters $\alpha_{SS}^{\ell,k}$, $\alpha_{SR}^{\ell,k}$, $\alpha_{RS}^{\ell,k}$ and $\alpha_{RR}^{\ell,k}$. This cross-stitch unit calculates $X_S'^{\ell,k} = \alpha_{SS}^{\ell,k} \cdot X_S^{\ell,k} + \alpha_{SR}^{\ell,k} \cdot X_R^{\ell,k}$ for the segmentation path and $X_R'^{\ell,k} = \alpha_{RS}^{\ell,k} \cdot X_S^{\ell,k} + \alpha_{RR}^{\ell,k} \cdot X_R^{\ell,k}$ for the registration path. The cross-stitch network has the advantage of being able to learn to strongly share feature maps between the tasks if that is beneficial. Conversely, if it is better for pairs of feature maps to be completely independent, the network can learn the identity matrix to separate those feature maps. This allows representations to be shared between the two paths in a flexible manner, at a negligible cost in terms of number of parameters.

We place the cross-stitch units after the downsampling and upsampling layers, so at four positions in total. This is in line with the original cross-stitch paper, where the authors suggest placing cross-stitch units after every pooling activation map. We found that the number of units is more crucial than their location as long as the units are distributed evenly across the network. For example, placing the cross-stitch units before the downsampling and upsampling layers instead of after them does not alter the performance, but placing a large number of cross-stitch units, such as units after every layer, will degrade the performance of the network. The proposed architecture is visualized in Figure 2.

## 3. Experiments and Results

### 3.1. Dataset

This study involves two different datasets from two different institutes and scanners, for patients who underwent intensity-modulated radiotherapy for prostate cancer. The first dataset is from Haukeland Medical Center (HMC), Norway. The dataset has 18 patients with 8-11 CT scans, each corresponding to a treatment fraction. These scans were acquired using a GE scanner, and have 90 to 180 slices with a voxel size of approximately $0.9 \times 0.9 \times 2.0$ mm. The second dataset is from Erasmus Medical Center (EMC), The Netherlands. This dataset consists of 14 patients with 3 follow-up CT scans each. The scans were acquired using a Siemens scanner, and have 91 to 218 slices with a voxel size of approximately $0.9 \times 0.9 \times 1.5$ mm. The target structures (prostate and seminal vesicles) as well as organs-at-risk (bladder and rectum) were manually delineated by radiation oncologists. The networks were trained and validated on the HMC dataset, while the EMC dataset was used as an independent test set. Training was performed on a subset of 111 image pairs from 12 patients, and validation was carried out on the remaining 50 image pairs from 6 patients. All datasets were resampled to an isotropic voxel size of $1 \times 1 \times 1$ mm.

### 3.2. Implementation and Training Details

We implemented the networks using TensorFlow (Abadi et al., 2016). The convolution layers were initialized from a random normal distribution with a mean of 0 and a standard deviation of 0.02, and the trainable alpha parameters of the cross-stitch units were initialized between 0 and 1 from a truncated random normal distribution with a mean of 0.5 and a standard deviation of 0.25. The number of filters was set to $\{16, 32, 64, 32, 16\}$ for the cross-stitch network and $\{23, 45, 91, 45, 23\}$ ($\sqrt{2}$ times as many) for the other networks in order to ensure that each network has approximately the same number of trainable parameters, namely $7.8 \cdot 10^5$. The patches were sampled equally from the organs-at-risk, the targets, and the remainder of the abdomen. We used the RAdam (Liu et al., 2019) optimizer with a learning rate of $10^{-4}$. The networks were trained for 200,000 iterations with an initial batch size of 2. In each batch, the training samples are doubled by switching the role of the fixed and moving patches, resulting in an effective batch size of four. The weights of the Dice and NCC losses were set to 1 and that of the bending energy loss to 0.5. For the total loss, all resolutions are weighted equally, namely $\frac{1}{3}$ each. Training, validation and testing were performed on a Nvidia GTX1080 Ti GPU with 11 GB of memory.

### 3.3. Evaluation Measures and Comparative Methods

The networks were evaluated in terms of their Mean Surface Distance (MSD) between the predicted segmentations and ground truth contours. The appendix contains results in terms of the DSC and the 95% Hausdorff Distance (HD).

We compare the proposed approach to three state-of-the-art methods in abdominal CT radiotherapy: one iterative method, one deep learning method and one hybrid method.

- **Elastix** (Qiao, 2017; Klein et al., 2010), a conventional iterative registration method. The Mutual Information similarity measure was used since it was found to perform better than

Table 1: MSD (mm) values for the different approaches on the HMC dataset. † denotes a significant difference (at $p = 0.05$) between the cross-stitch network and the other networks.

| | Output Path | Prostate | | Seminal vesicles | | Rectum | | Bladder | |
|---|---|---|---|---|---|---|---|---|---|
| | | $\mu \pm \sigma$ | Median | $\mu \pm \sigma$ | Median | $\mu \pm \sigma$ | Median | $\mu \pm \sigma$ | Median |
| Segmentation | | $1.49 \pm 0.3^{\dagger}$ | 1.49 | $2.50 \pm 2.6^{\dagger}$ | 2.09 | $3.39 \pm 2.2^{\dagger}$ | 2.73 | $1.60 \pm 1.1^{\dagger}$ | 1.13 |
| Registration | | $1.43 \pm 0.8^{\dagger}$ | 1.29 | $1.71 \pm 1.4^{\dagger}$ | 1.37 | $2.44 \pm 1.1^{\dagger}$ | 2.17 | $3.40 \pm 2.3^{\dagger}$ | 2.71 |
| JRS-Registration | | $1.20 \pm 0.4^{\dagger}$ | 1.13 | $1.35 \pm 0.7$ | 1.16 | $2.08 \pm 1.0^{\dagger}$ | 1.82 | $2.63 \pm 2.3^{\dagger}$ | 1.90 |
| Fully Hard Sharing | *Segmentation* | $1.14 \pm 0.4^{\dagger}$ | 1.06 | $1.73 \pm 2.1$ | 1.12 | $1.91 \pm 0.9$ | 1.64 | $1.04 \pm 0.7^{\dagger}$ | 0.87 |
| | *Registration* | $1.20 \pm 0.3^{\dagger}$ | 1.11 | $1.33 \pm 0.7$ | **1.10** | $2.16 \pm 1.1^{\dagger}$ | 1.85 | $2.56 \pm 1.9^{\dagger}$ | 1.90 |
| Cross-Stitch | *Segmentation* | $\mathbf{1.06 \pm 0.3}$ | **0.99** | $\mathbf{1.27 \pm 0.4}$ | 1.15 | $\mathbf{1.76 \pm 0.8}$ | **1.47** | $\mathbf{0.91 \pm 0.4}$ | **0.82** |
| | *Registration* | $1.10 \pm 0.3$ | 1.06 | $1.30 \pm 0.6$ | 1.13 | $2.00 \pm 1.0$ | 1.75 | $2.45 \pm 2.1$ | 1.81 |
| Elastix (Qiao, 2017) | | $1.73 \pm 0.7^{\dagger}$ | 1.59 | $2.71 \pm 1.6^{\dagger}$ | 2.45 | $3.69 \pm 1.2^{\dagger}$ | 3.50 | $5.26 \pm 2.6^{\dagger}$ | 4.72 |
| JRS-GAN (Elmahdy et al., 2019b) | | $1.14 \pm 0.3^{\dagger}$ | 1.04 | $1.75 \pm 1.3^{\dagger}$ | 1.44 | $2.17 \pm 1.1^{\dagger}$ | 1.89 | $2.25 \pm 1.9^{\dagger}$ | 1.54 |
| Hybrid (Elmahdy et al., 2019a) | | $1.27 \pm 0.3^{\dagger}$ | 1.25 | $1.47 \pm 0.5^{\dagger}$ | 1.32 | $2.03 \pm 0.6^{\dagger}$ | 1.85 | $1.75 \pm 1.0^{\dagger}$ | 1.26 |

the NCC similarity measure on the validation set. The transformation is parameterized by B-splines.

- **JRS-GAN** (Elmahdy et al., 2019b), a deep learning approach that trains a registration network for contour propagation with a joint loss similar to our JRS-registration network, and a discriminator network for giving feedback on the warped images and contours.
- **Hybrid** (Elmahdy et al., 2019a), a hybrid learning and iterative approach. A CNN network segments the bladder and feeds it to the registration model as prior knowledge of the underlying anatomy. It integrates domain-specific strategies such as gas pocket inpainting, contrast clipping and focused registration for the seminal vesicles and rectum.

The inference speed is less than a second for the deep learning methods, and in the order of minutes for the iterative and hybrid approaches.

### 3.4. Evaluation of Architectures on the HMC Dataset

Quantitative results are given in Table 1, and example results in Figure 3. The first two rows in Table 1 show the results from the single-task networks in terms of MSD. The registration network works better than the segmentation network on most organs as it essentially uses prior knowledge of the organs of the patient by warping the manually delineated planning scan. The segmentation network performed better on the bladder, since the registration network often had trouble establishing a correspondence between the bladder in the fixed image and the moving image as this organ tends to deform considerably between visits. The segmentation network failed to classify any voxel as seminal vesicles in 5 cases. The seminal vesicles are hard to identify because of their small size and poor contrast, which explains the relatively poor performance of the segmentation network on this organ. The registration network has the benefit of being able to use the context, namely the surrounding anatomical features and organs, to more accurately warp the seminal vesicles into place.

The results from the loss-joined JRS-registration network are shown in the third row of Table 1. It is clear that the additional segmentation loss during training improves the registration quality significantly.

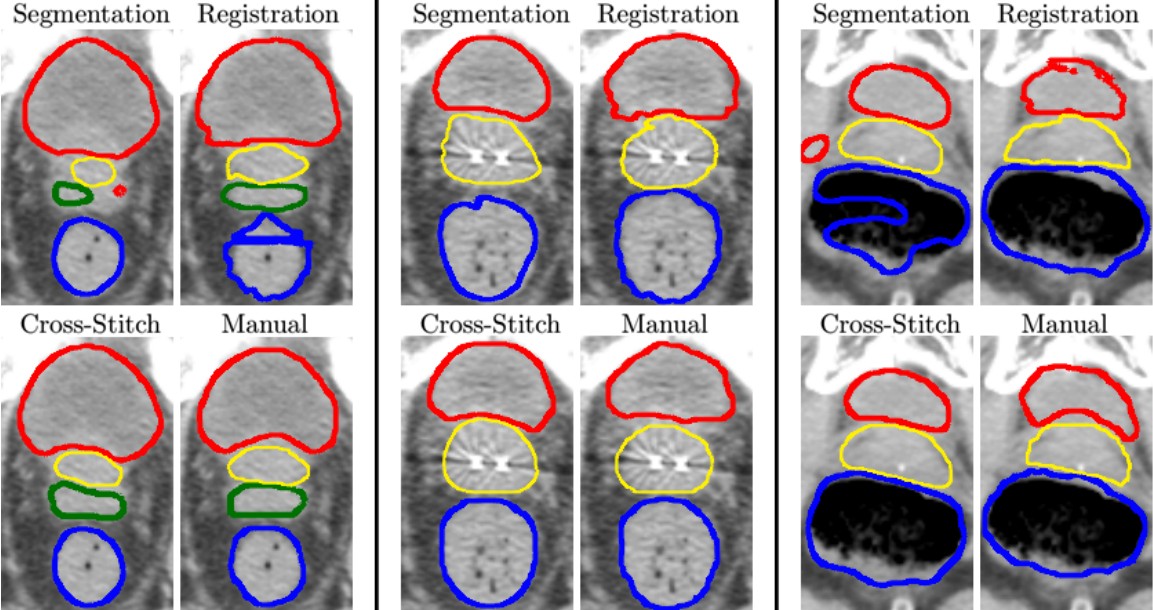

Figure 3: Example contours generated by the single-task networks and the cross-stitch network on the HMC dataset. From left to right, the selected cases are the first, second and third quantile in terms of prostate MSD of the cross-stitch network.

The fourth and fifth rows in Table 1 show the results of the fully-hard parameter sharing network. The contours from its segmentation path see substantial improvements in accuracy over the contours from the segmentation network. The registration path yields improvements over the single-task registration network, but it does not improve over the JRS-registration network. These results demonstrate that architecturally joining segmentation and registration can be very beneficial for the segmentation output and can yield more accurate segmentations than either of the single-task networks.

The cross-stitch network performs the best of all networks, as demonstrated by the results in Table 1. Both the segmentation path and the registration path improve over the corresponding paths of the hard parameter sharing network, though it is again the segmentation path that typically yields the most accurate contours. The proposed joint networks, particularly the cross-stitch network, yield significantly better contours than any of the state-of-the-art methods. These results confirm the effectiveness of architecturally joining registration and segmentation for generating accurate organ delineations.

### 3.5. Evaluation on the Independent EMC Test Set

Table 2 provides quantitative results on the independent test set. The segmentation network failed to classify any voxel as seminal vesicles in 5 cases, and the segmentation paths of the fully hard sharing network and the cross-stitch network in 1 case. Note that the deep-learning approaches have not been re-trained nor fine-tuned. Again, the joint networks

Table 2: MSD (mm) values for the different approaches on the independent EMC test set. † denotes a significant difference (at $p = 0.05$) between the cross-stitch network and the other networks. Results for JRS-GAN are not available for this dataset.

| | Output Path | Prostate | | Seminal vesicles | | Rectum | | Bladder | |
|---|---|---|---|---|---|---|---|---|---|
| | | $\mu \pm \sigma$ | Median | $\mu \pm \sigma$ | Median | $\mu \pm \sigma$ | Median | $\mu \pm \sigma$ | Median |
| Segmentation | | $3.18 \pm 1.8^{\dagger}$ | 2.57 | $9.33 \pm 10.1^{\dagger}$ | 5.82 | $5.79 \pm 3.4^{\dagger}$ | 5.18 | $\mathbf{1.88 \pm 1.5}$ | 1.50 |
| Registration | | $2.01 \pm 2.5^{\dagger}$ | 1.18 | $2.86 \pm 5.2^{\dagger}$ | 1.18 | $2.89 \pm 2.5^{\dagger}$ | 2.23 | $5.98 \pm 4.7^{\dagger}$ | 4.44 |
| JRS-Registration | | $1.96 \pm 2.6^{\dagger}$ | 1.16 | $2.60 \pm 4.9^{\dagger}$ | 1.07 | $2.64 \pm 2.3$ | 2.14 | $5.15 \pm 4.4^{\dagger}$ | 3.14 |
| Fully Hard Sharing | *Segmentation* | $2.02 \pm 2.5^{\dagger}$ | 1.34 | $6.34 \pm 10.3^{\dagger}$ | 1.98 | $3.27 \pm 2.9$ | $\mathbf{2.10}$ | $2.66 \pm 2.6^{\dagger}$ | 1.38 |
| | *Registration* | $2.00 \pm 2.6^{\dagger}$ | 1.20 | $2.66 \pm 5.2^{\dagger}$ | 1.12 | $2.66 \pm 2.2^{\dagger}$ | 2.24 | $5.09 \pm 4.2^{\dagger}$ | 2.84 |
| Cross-Stitch | *Segmentation* | $1.88 \pm 2.2$ | 1.21 | $4.73 \pm 8.0$ | 1.42 | $3.61 \pm 5.0$ | 2.18 | $2.45 \pm 2.4$ | $\mathbf{1.24}$ |
| | *Registration* | $1.82 \pm 2.4$ | $\mathbf{1.09}$ | $2.45 \pm 5.0$ | $\mathbf{1.02}$ | $\mathbf{2.57 \pm 2.3}$ | $\mathbf{2.10}$ | $4.93 \pm 4.1$ | 2.69 |
| Elastix (Qiao, 2017) | | $\mathbf{1.42 \pm 0.7}$ | 1.17 | $2.07 \pm 2.6^{\dagger}$ | 1.24 | $3.20 \pm 1.6^{\dagger}$ | 3.07 | $5.30 \pm 5.1^{\dagger}$ | 3.27 |
| Hybrid (Elmahdy et al., 2019a) | | $1.55 \pm 0.6^{\dagger}$ | 1.36 | $\mathbf{1.65 \pm 1.3}$ | 1.22 | $2.65 \pm 1.6$ | 2.36 | $3.81 \pm 3.6^{\dagger}$ | 2.26 |

outperform the single-task networks as well as the state-of-the art methods in terms of the median values that are less influenced by outliers. The mean values are relatively high compared to the median values. This discrepancy can be explained by the intensity variations between the population of the training set and test set causing more outliers.

## 4. Discussion and Conclusion

In this work, we proposed to architecturally join image registration and segmentation to generate daily organ delineations essential for adaptive image-guided radiotherapy. We experimented with different ways of intertwining registration and segmentation in three-dimensional fully convolutional neural networks, and found that joining the tasks with cross-stitch units works best. Via the cross-stitch units the network learns to exchange information between its registration path and segmentation path. Moreover, we studied the potential bias of the segmentation network by adding $S_m$, via an experiment for the single-task network where $S_m$ is fed to it alongside $I_f$. The segmentation network improved over feeding $I_f$ only, however it was still inferior to the cross-stitch network, and therefore it was not included in this paper. The segmentation network without $S_m$ was included instead as it serves as a vanilla segmentation baseline.

Evaluation on a validation set and an independent test set demonstrated that the proposed joining of segmentation and registration significantly outperforms their single-task counterparts. On the validation set the proposed approach outperformed existing methods, sometimes by a margin. On the independent test set existing methods achieved better mean values for the prostate and seminal vesicles, while for the rectum the proposed methods performed better. For the bladder specifically, the single-task segmentation network achieved better mean values than the other networks due to the fact that for the HMC dataset, which was used for training and validation, a bladder filling protocol was in place, meaning that the deformation of the bladder between different visits and planning is not large. However, this is not the case for the EMC dataset, the test set. Since the registration-based networks and joint networks were trained on small bladder deformations, they had trouble with larger deformations. The segmentation network was not affected since it does not depend on the deformation but rather on the underlying texture to segment the bladder. This issue could

be relatively easily addressed by including synthetic larger deformations during training or including a few patients from the EMC dataset into the training. Nevertheless, in terms of median values, the proposed method was superior for all the organs even though we did not use domain-specific strategies similar to the ones presented in (Elmahdy et al., 2019a). Retraining or fine-tuning for this patient population and scanner type can further improve the results for the proposed methods.

A promising direction for future research is to investigate the addition of a third task to the joint networks, notably the generation of the radiotherapy treatment plan. This may allow the joint networks to generate delineations with favorable dosimetric features. Further investigations could be towards the generalization of the network across different patient populations and scanners. Finally, we hypothesize that the accuracy of the networks could be further improved by including more organs, such as the lymph nodes, as this provides extra guidance.

In conclusion, on the validation set and the independent test set, the proposed approach yielded median mean surface distances around the slice thickness. Our approach achieved 0.99 mm, 0.82 mm, 1.13 mm, and 1.47 mm on the validation set and 1.09 mm, 1.24 mm, 1.02 mm, and 2.10 mm on the test set for the prostate, bladder, seminal vesicles, and rectum, respectively. With an inference speed of less than a second, our approach is ideal for generating the daily contours in online adaptive image-guided radiotherapy, and subsequently reducing treatment-related side effects and improve quality-of-life for patients after treatment.

## Acknowledgments

The HMC dataset with contours was collected at Haukeland University Hospital, Bergen, Norway, and was provided to us by responsible oncologist Svein Inge Helle and physicist Liv Bolstad Hysing. The EMC dataset with contours was collected at Erasmus University Medical Center, Rotterdam, The Netherlands, and was provided to us by radiation therapist Luca Incrocci and physicist Mischa Hoogeman. They are gratefully acknowledged.

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

## Appendix A. of the paper "A Cross-Stitch Architecture for Joint Registration and Segmentation in Adaptive Radiotherapy"

In this appendix we highlight the details of the network architecture as well as detailed results of the proposed method.

### A.1. Base Network Architecture

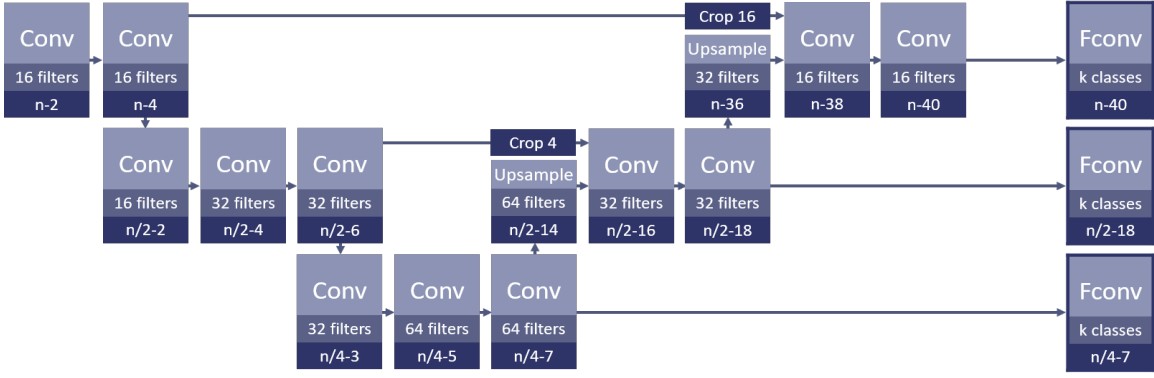

Figure 4: The base architecture used for our networks, a three-dimensional deep convolutional neural network derived from U-Net (Ronneberger et al., 2015), inspired by (Fan et al., 2018). The number of filters and output sizes are shown below each layer. The output of each layer is cubic, as the input patches are of size $n^3$.

### A.2. Experimental Results in Terms of DSC and 95%HD

Table 3: DSC values for the different approaches on the HMC dataset.

| | | Prostate | | Seminal vesicles | | Rectum | | Bladder | |
|---|---|---|---|---|---|---|---|---|---|
| | Output Path | $\mu \pm \sigma$ | Median | $\mu \pm \sigma$ | Median | $\mu \pm \sigma$ | Median | $\mu \pm \sigma$ | Median |
| Segmentation | | $0.84 \pm 0.03^{\dagger}$ | 0.84 | $0.60 \pm 0.14^{\dagger}$ | 0.62 | $0.75 \pm 0.10^{\dagger}$ | 0.77 | $0.90 \pm 0.07^{\dagger}$ | 0.93 |
| Registration | | $0.85 \pm 0.06^{\dagger}$ | 0.86 | $0.62 \pm 0.18^{\dagger}$ | 0.68 | $0.79 \pm 0.08^{\dagger}$ | 0.81 | $0.82 \pm 0.10^{\dagger}$ | 0.84 |
| JRS-Registration | | $0.87 \pm 0.04^{\dagger}$ | 0.87 | $0.67 \pm 0.15^{\dagger}$ | 0.72 | $0.83 \pm 0.06^{\dagger}$ | 0.84 | $0.87 \pm 0.08^{\dagger}$ | 0.91 |
| Fully Hard Sharing | *Segmentation* | $0.87 \pm 0.05^{\dagger}$ | 0.88 | $\mathbf{0.70 \pm 0.11}$ | $\mathbf{0.74}$ | $0.84 \pm 0.06$ | 0.86 | $\mathbf{0.94 \pm 0.03}^{\dagger}$ | $\mathbf{0.95}$ |
| | *Registration* | $0.86 \pm 0.04^{\dagger}$ | 0.87 | $0.68 \pm 0.15^{\dagger}$ | 0.72 | $0.82 \pm 0.06^{\dagger}$ | 0.84 | $0.87 \pm 0.07^{\dagger}$ | 0.90 |
| Cross-Stitch | *Segmentation* | $\mathbf{0.88 \pm 0.04}$ | 0.88 | $\mathbf{0.70 \pm 0.11}$ | $\mathbf{0.74}$ | $\mathbf{0.86 \pm 0.05}$ | $\mathbf{0.88}$ | $\mathbf{0.94 \pm 0.02}$ | $\mathbf{0.95}$ |
| | *Registration* | $0.87 \pm 0.03$ | 0.88 | $0.68 \pm 0.15$ | 0.73 | $0.84 \pm 0.05$ | 0.85 | $0.88 \pm 0.08$ | 0.91 |
| Elastix (Qiao, 2017) | | $0.84 \pm 0.07^{\dagger}$ | 0.86 | $0.50 \pm 0.25^{\dagger}$ | 0.53 | $0.74 \pm 0.06^{\dagger}$ | 0.74 | $0.75 \pm 0.10^{\dagger}$ | 0.76 |
| JRS-GAN (Elmahdy et al., 2019b) | | $0.86 \pm 0.04^{\dagger}$ | 0.87 | $0.61 \pm 0.20^{\dagger}$ | 0.67 | $0.82 \pm 0.06^{\dagger}$ | 0.83 | $0.88 \pm 0.08^{\dagger}$ | 0.92 |
| Hybrid (Elmahdy et al., 2019a) | | $\mathbf{0.88 \pm 0.04}$ | $\mathbf{0.89}$ | $\mathbf{0.70 \pm 0.14}$ | 0.72 | $0.85 \pm 0.06$ | 0.87 | $0.91 \pm 0.08^{\dagger}$ | $\mathbf{0.95}$ |

Table 4: DSC values for the different approaches on the independent EMC test set.

| | Output Path | Prostate $\mu \pm \sigma$ | Median | Seminal vesicles $\mu \pm \sigma$ | Median | Rectum $\mu \pm \sigma$ | Median | Bladder $\mu \pm \sigma$ | Median |
|---|---|---|---|---|---|---|---|---|---|
| Segmentation | | $0.73 \pm 0.11^{\dagger}$ | 0.77 | $0.37 \pm 0.30^{\dagger}$ | 0.28 | $0.67 \pm 0.10^{\dagger}$ | 0.68 | $\mathbf{0.91 \pm 0.07}$ | **0.93** |
| Registration | | $0.83 \pm 0.16$ | 0.88 | $0.64 \pm 0.26^{\dagger}$ | 0.74 | $0.72 \pm 0.16^{\dagger}$ | 0.77 | $0.75 \pm 0.19^{\dagger}$ | 0.82 |
| JRS-Registration | | $0.84 \pm 0.16$ | 0.89 | $0.67 \pm 0.25$ | 0.79 | $0.76 \pm 0.14^{\dagger}$ | 0.79 | $0.79 \pm 0.17^{\dagger}$ | 0.88 |
| Fully Hard Sharing | *Segmentation* | $0.83 \pm 0.15^{\dagger}$ | 0.88 | $0.55 \pm 0.29^{\dagger}$ | 0.65 | $0.78 \pm 0.16$ | 0.81 | $0.88 \pm 0.11$ | **0.93** |
| | *Registration* | $0.83 \pm 0.16^{\dagger}$ | 0.88 | $0.66 \pm 0.25^{\dagger}$ | 0.75 | $0.76 \pm 0.15^{\dagger}$ | 0.80 | $0.79 \pm 0.16^{\dagger}$ | 0.87 |
| Cross-Stitch | *Segmentation* | $0.84 \pm 0.14$ | 0.89 | $0.61 \pm 0.27$ | 0.73 | $0.78 \pm 0.14$ | 0.81 | $0.88 \pm 0.10$ | **0.93** |
| | *Registration* | $0.84 \pm 0.15$ | 0.89 | $0.68 \pm 0.24$ | 0.80 | $0.77 \pm 0.15$ | 0.80 | $0.80 \pm 0.16$ | 0.87 |
| Elastix (Qiao, 2017) | | $\mathbf{0.89 \pm 0.05}^{\dagger}$ | **0.91** | $0.72 \pm 0.24$ | **0.82** | $0.75 \pm 0.12^{\dagger}$ | 0.76 | $0.79 \pm 0.18^{\dagger}$ | 0.87 |
| Hybrid (Elmahdy et al., 2019a) | | $0.88 \pm 0.04$ | 0.89 | $\mathbf{0.77 \pm 0.15}^{\dagger}$ | 0.81 | $\mathbf{0.80 \pm 0.10}$ | **0.82** | $0.85 \pm 0.13^{\dagger}$ | 0.90 |

Table 5: 95%HD values for the different approaches on the HMC dataset.

| | Output Path | Prostate $\mu \pm \sigma$ | Median | Seminal vesicles $\mu \pm \sigma$ | Median | Rectum $\mu \pm \sigma$ | Median | Bladder $\mu \pm \sigma$ | Median |
|---|---|---|---|---|---|---|---|---|---|
| Segmentation | | $4.4 \pm 1.0^{\dagger}$ | 4.4 | $8.6 \pm 8.6^{\dagger}$ | 7.3 | $16.5 \pm 11.0^{\dagger}$ | 13.3 | $6.9 \pm 6.6^{\dagger}$ | 4.0 |
| Registration | | $5.5 \pm 4.5^{\dagger}$ | 4.0 | $5.6 \pm 4.1^{\dagger}$ | 4.3 | $11.0 \pm 6.4^{\dagger}$ | 9.4 | $15.7 \pm 9.6^{\dagger}$ | 12.1 |
| JRS-Registration | | $3.6 \pm 1.9^{\dagger}$ | 3.1 | $4.4 \pm 2.8$ | 3.7 | $9.8 \pm 5.9$ | 8.1 | $13.4 \pm 10.7^{\dagger}$ | 10.6 |
| Fully Hard Sharing | *Segmentation* | $3.4 \pm 1.2^{\dagger}$ | 3.0 | $6.3 \pm 9.4$ | 3.6 | $10.1 \pm 5.7$ | 8.9 | $3.9 \pm 4.8^{\dagger}$ | 3.0 |
| | *Registration* | $3.7 \pm 1.4^{\dagger}$ | 3.2 | $4.5 \pm 3.3$ | 3.2 | $10.4 \pm 6.1$ | 9.1 | $12.7 \pm 9.6^{\dagger}$ | 9.7 |
| Cross-Stitch | *Segmentation* | $3.0 \pm 1.0$ | 3.0 | $4.3 \pm 1.7$ | 3.9 | $9.5 \pm 6.2$ | 7.2 | $\mathbf{3.3 \pm 2.9}$ | **2.3** |
| | *Registration* | $3.2 \pm 0.9$ | 3.0 | $4.5 \pm 3.3$ | 3.6 | $9.8 \pm 6.3$ | 8.6 | $12.2 \pm 10.1$ | 9.7 |
| Elastix (Qiao, 2017) | | $4.0 \pm 1.7^{\dagger}$ | 3.7 | $6.0 \pm 3.4^{\dagger}$ | 5.6 | $10.9 \pm 5.2^{\dagger}$ | 9.8 | $15.3 \pm 8.3^{\dagger}$ | 13.6 |
| JRS-GAN (Elmahdy et al., 2019b) | | $3.4 \pm 1.2^{\dagger}$ | 3.0 | $5.3 \pm 3.0^{\dagger}$ | 4.6 | $10.1 \pm 6.1$ | 8.4 | $11.0 \pm 9.6^{\dagger}$ | 7.6 |
| Hybrid (Elmahdy et al., 2019a) | | $\mathbf{2.9 \pm 0.9}$ | **2.8** | $\mathbf{3.8 \pm 2.2}^{\dagger}$ | **3.1** | $\mathbf{7.7 \pm 4.5}$ | **6.1** | $5.7 \pm 4.6^{\dagger}$ | 3.3 |

Table 6: 95%HD values for the different approaches on the independent EMC test set.

| | Output Path | Prostate $\mu \pm \sigma$ | Median | Seminal vesicles $\mu \pm \sigma$ | Median | Rectum $\mu \pm \sigma$ | Median | Bladder $\mu \pm \sigma$ | Median |
|---|---|---|---|---|---|---|---|---|---|
| Segmentation | | $10.7 \pm 5.4^{\dagger}$ | 9.3 | $21.4 \pm 17.9^{\dagger}$ | 15.4 | $30.5 \pm 12.9^{\dagger}$ | 29.0 | $11.2 \pm 8.5$ | 10.0 |
| Registration | | $6.7 \pm 5.9^{\dagger}$ | 4.2 | $7.5 \pm 8.6^{\dagger}$ | 4.3 | $13.1 \pm 6.9^{\dagger}$ | 12.0 | $22.7 \pm 14.0^{\dagger}$ | 20.2 |
| JRS-Registration | | $5.2 \pm 5.7$ | 3.2 | $6.5 \pm 7.1^{\dagger}$ | 4.0 | $12.6 \pm 6.7$ | 12.0 | $20.3 \pm 14.0^{\dagger}$ | 18.6 |
| Fully Hard Sharing | *Segmentation* | $5.7 \pm 5.4^{\dagger}$ | 4.1 | $14.4 \pm 17.2^{\dagger}$ | 6.8 | $16.8 \pm 12.6$ | 13.6 | $10.9 \pm 10.9^{\dagger}$ | 5.5 |
| | *Registration* | $5.6 \pm 5.6^{\dagger}$ | 4.0 | $6.6 \pm 7.8^{\dagger}$ | 4.0 | $13.1 \pm 6.7^{\dagger}$ | 13.0 | $19.6 \pm 12.0^{\dagger}$ | 17.4 |
| Cross-Stitch | *Segmentation* | $5.8 \pm 5.4$ | 4.0 | $12.2 \pm 15.8$ | 5.0 | $17.0 \pm 14.7$ | 14.0 | $\mathbf{10.8 \pm 11.3}$ | **4.4** |
| | *Registration* | $5.1 \pm 5.5$ | 3.2 | $6.2 \pm 8.6$ | 3.3 | $12.6 \pm 6.7$ | 12.0 | $19.1 \pm 12.5$ | 16.2 |
| Elastix (Qiao, 2017) | | $\mathbf{3.6 \pm 2.0}$ | **2.9** | $\mathbf{4.6 \pm 4.4}$ | 3.2 | $11.3 \pm 6.0$ | 11.3 | $16.1 \pm 14.8^{\dagger}$ | 10.4 |
| Hybrid (Elmahdy et al., 2019a) | | $3.9 \pm 1.9$ | 3.4 | $4.8 \pm 4.7$ | **3.1** | $\mathbf{10.3 \pm 6.8}^{\dagger}$ | **8.6** | $11.1 \pm 10.6$ | 6.6 |

