# OpenReview forum: "A Cross-Stitch Architecture for Joint Registration and Segmentation in Adaptive Radiotherapy"
_MIDL.io/2020/Conference — MIDL 2020_

### Official Review · AnonReviewer4 · 2020-03-01
**The authors propose a joint registration segmentation method that shows promising results and is well evaluated.**

**Rating:** 3
**Confidence:** 5
**Recommendation:** Oral

**Summary:**

The authors propose a new network architecture to jointly learn multiple task (segmentation and registration) by using cross-stitch units. They perform a detailed evaluation of  their method on the task of recontouring in adaptive radiotherapy (CT-CT follow-up registration/propagation). The proposed method shows good results on the dataset on that the network was trained on. However, on an independent dataset the performance is less good.

**Strengths:**

•	The authors present a new method for jointly learn registration and segmentation by introducing a new architecture that combines both tasks in one network.
•	They show that combining multiple tasks (segmentation and registration) can help to increase the performs of both tasks.
•	The authors perform a detailed evaluation by comparing their method with:
          * a single-task registration network
          * a single-task segmentation network
          * a registration network which includes the segmentation in the loss function
          * a joint registration segmentation network using hard parameter sharing
          * Elastix
          * a recent deep-learning method which uses a discriminator network for giving feedback on the warped images and contours.
•	The authors are aware that a different number of filters in the used architectures might falsify the result. They took care of that by adapting the number of filters for each architecture so that all networks approximately have the same number of filters.
•	The authors show the transferability of their trained network by evaluating it on an independent dataset which wasn’t used during training. However, the results are less good on this dataset.


**Weaknesses:**

•	The authors don’t describe if and how the deformation field patches are combined into an overall deformation field. Without this, the proposed method is more useful for the segmentation task rather than the registration task. Especially for the dose accumulation, a full deformation field is needed.
•	The authors don’t evaluate the number of foldings. However, this is an important measure to evaluate registration quality!
•	There are still a number of unanswered questions.


**Detailed Comments:**

•	I am curious if there are any feature maps for which it is better to be completely independent and therefore are not shared between both tasks. Can you please comment on that?
•	How much does deep supervision improve the result?

**Justification Of Rating:**

The authors propose an interesting new approach for joint segmentation and registration with an detailed evaluation. Therefore this work should be presented at MIDL. However, there are still some questions open that should be answered.

**Paper Type:**

both

**Questions To Address In The Rebuttal:**

•	Are the images preprocessed? The authors mentioned that they are affinely aligned beforehand, but they don’t say if there are any further preprocessing steps (e.g. resampling, intensity scaling) like in the cited paper that also used this data.
•	How big is the patch size?
•	Are the calculated deformation fields of each patch are combined to an overall deformation field? If yes, how are the deformation field be combined?
•	Do any foldings occur in the deformation field?




**Special Issue:**

no

---

> ### Author Response · Authors · 2020-03-27
> **Answer to Reviewer 4**
>
> We would like to thank the reviewer for the positive feedback on paper novelty, clarity and robustness of the results across different vendors.
>
> Regarding the preprocessing of the images, all the data were resampled to an isotropic voxel size of 1 $\times$ 1 $\times$ 1 mm. We updated the manuscript to explain so.
>
> Regarding the patch size, We chose a patch size of 96 $\times$ 96 $\times$ 96. We added the patch size to the revised version.
>
> Regarding the generation of the overall deformation field, we extracted overlapping patches and estimated the DVF, and subsequently stitch the estimated DVFs.
>
> Regarding the preference of the network for sharing feature maps, we noticed that the network prefers to share representation in the contracting path, while the opposite in the expansive path.
>
> For the effectiveness of deep supervision, we believe that deep supervision is important because at the lower resolutions the network can focus on large organs or large deformations and vice versa at higher resolutions. Here, we rely on the original paper [1], who demonstrated the benefits of deep supervision, which was also repeatedly confirmed in the literature. We added this information to Section 2.1. However, it would be interesting to confirm, by an additional experiment in the journal version.
>
> [1] https://arxiv.org/abs/1709.00799

---

> > ### Comment · AnonReviewer4 · 2020-04-01
> > **Responses**
> >
> > Thank you for the clarifications for my questions.
> >
> > Could you please answer also my last question whether any foldings occur in the deformation field?

---

> > > ### Author Response · Authors · 2020-04-02
> > > **Answer to Reviewer 4**
> > >
> > > We did not visually observe foldings via the deformed images, but will study this further and quantify this in the journal version of the work.

---

### Official Review · AnonReviewer1 · 2020-03-12
**Nice results showcasing the application of Cross-Stitch modules for radiotherapy but more experiments would improve the quality of the paper**

**Rating:** 3
**Confidence:** 5
**Recommendation:** Poster

**Summary:**

The authors applied principles of multi-task learning towards adaptive radiotherapy. The authors investigated the use of Cross-Stitch modules to improve the performance of organ at risk (OAR) segmentation and the registration of CT scans. The authors demonstrated that a multi-task network trained with Cross-Stitch performed best when validated on data similar to the training distributions. The authors also show that the Cross-Stitch method outperforms single task and vanilla hard-parameter sharing on the independent validation set whilst being competitive with respect to domain-specific strategies. The authors have successfully demonstrated the applicability of training a 3D multi-task network for image-guided radiation therapy.

**Strengths:**

[1] The paper is well written and it is mainly clear what the experiments were in addition to the main observations

[2] The quantitative results are very good and the qualitative results nicely illustrate the benefit of training tasks in a multi-task setting.

[3] The results on the independent test set demonstrate that the Cross-Stitch network can remain competitive when trained on datasets stemming from a different medical centre. It is interesting to see that the performance of single-task networks drops dramatically but the cross-stitch and hard-parameter sharing networks still perform adequately.

**Weaknesses:**

* In my opinion, this is mainly an application paper as there is no new methodology presented. However, for an application paper, I find that the experiments are lacking. It is known in the literature that multi-task learning is likely to improve performance of tasks in comparison to single task networks. If the main objective was to showcase that multi-task learning can improve segmentation and registration tasks for radiotherapy, then various multi-task learning methods should have been compared with discussions centred upon which methods can benefit radiotherapy the most. There are numerous other techniques, which could additionally have been tested such as: multi-task learning with homoscedastic task uncertainty [1], with heteroscedastic uncertainty [2] and with gradient normalisation of task gradients [3].

[1] Kendall et al. https://arxiv.org/abs/1705.07115
[2] Bragman et al. https://arxiv.org/abs/1806.06595
[3] Chen et al. - https://arxiv.org/abs/1711.02257


**Detailed Comments:**

[1] It would be best to enumerate which concepts from multi-task learning you are exploiting and citing the relevant literature instead of pointing to Ruder's blog post on multi-task learning.

[2] What was the inference time of other techniques?

**Justification Of Rating:**

The authors have demonstrated well the capabilities of multi-task learning for joint learning of image segmentation and image registration for adaptive radiotherapy. The authors demonstrated that using Cross-Stitch modules produces the best results when compared with single task networks, hard-parameter sharing networks and other methods. The results are good and demonstrate that multi-task learning should be used. However, I find the experiments lacking for an application paper with no real, new insights; either about the analysed methods or the clinical applicability.

**Paper Type:**

validation/application paper

**Questions To Address In The Rebuttal:**

* It has been shown in multiple multi-task learning papers [1, 2] and in the Cross-Stitch paper that vanilla hard-parameter sharing methods are not entirely likely to outperform both single task networks (even though you were able to outperform both with the fully hard-sharing method in Table 1).

Were you able to evaluate whether methods that dynamically weight the loss functions in the multi-task objective: a) perform better than fully hard sharing and b) are competitive with a Cross-Stitch network? Furthermore, unlike Cross-Stitch, these methods do not require learning to share between 2 trained single task networks and therefore use at less parameters.

* For the results in Table 1, the authors trained your networks on 12 patients and report validation metrics on the following 6 patients. Was any cross-validation performed on the 12 patients? How did one balance training and overfitting?

* Is there any particular reason why the performance for Bladder segmentation in Table 2 was best for single-task network?

* Was your Cross-Stitch network trained end-to-end with the Cross-Stitch modules or was it trained sequentially? If it was trained end-to-end, how do you expect this to have affected the learned representations? Do you expect a sequentially trained Cross-Stitch network to perform better?

* In the multi-task architecture in Figure 2, the segmentation network receives as input the daily CT scan (I_f), the registration network receives the planning scan (I_m), the daily scan (I_f) and the segmentation S_m. Since you are training cross-stitch networks, to share information, how likely is it that the segmentation network's performance is biased by the extra segmentation information obtained by S_m? Does this mean one cannot compare the segmentation results between the multi-task networks and the single task segmentation network?

[1] Kendall et al. https://arxiv.org/abs/1705.07115
[2] Bragman et al. https://arxiv.org/abs/1806.06595

**Special Issue:**

no

---

> ### Author Response · Authors · 2020-03-27
> **Answer to Reviewer 1**
>
> We would like to thank the reviewer for the positive feedback on paper clarity and robustness of the results across different vendors.
>
> Regarding the superiority of the fully hard sharing network over the single-task networks, we believe it is because image registration and segmentation are highly correlated tasks. Therefore, we believe the tasks can use each other's intermediate predictions inside the network for guidance, which improves the accuracy of the network over the single-task networks.
>
> Regarding the dynamic loss weighting, in this paper, we fixed the loss weights and focused on studying the performance of different architectures derived from or related to cross-stitch methodology. Moreover, we compared it with (Elmahdy et. al., 2019b) where the loss is partially defined by the discriminator network and the cross-stitch network surpassed this method. However, it would indeed be interesting to compare all the networks presented in the paper with and without dynamic loss weighting to explore its effect. We would like to point out that the method was compared against quite a number of baseline methods. It is certainly possible to compare against more, and the mentioned ones are indeed interesting, but in our opinion not in the scope and length of a conference paper. Regarding the number of parameters, the overhead of cross-stitch modules is very minimal since it only adds 4 learnable parameters per feature map, which is negligible compared to the overall number of parameters.
>
> For cross-validation, in this version of the paper we did not perform cross-validation due to the limited computational resources given the large number of experiments. Instead we tested all the networks on an independent test set from a different vendor and hospital than the training and validation sets. In the journal version we indeed plan to add cross-validation.
>
> Regarding the performance of bladder segmentation in Table 2, for the Haukeland dataset, which was used for training and validation, a bladder filling protocol was in place, meaning that the deformation of the bladder between different visits and planning is not large. However, this is not the scenario for Erasmus dataset, the test set. Since the registration-based networks and joint networks were trained on small bladder deformations, it had trouble with larger deformations. The segmentation network was not affected since it does not depend on the deformation but rather on the underlying texture to segment the bladder. This issue could be relatively easily addressed by including synthetic larger deformations during training or include few patients from the EMC dataset into the training. We added this observation to the Discussion.
>
> Regarding the training of the cross-stitch network, it was trained end-to-end. In our opinion, training the network end-to-end would give the network more freedom to either learn shared representative features or task specific features. Moreover, we initialized the cross-stitch weights with truncated normally distributed weights with mean 0.5 to encourage the network to share representations. If you mean by sequential training, initializing the network with pre-trained task specific weights, we think that might bias the network towards a certain task over the other, it also might lead one or both networks to reach a local optimum, where the cross-stitch network might not be able to optimally share representations across the tasks.
>
> Regarding the potential bias of the segmentation network by $S_m$, we actually had the same thought, so we previously run an experiment for the single-task network while feeding $S_m$ alongside $I_f$. The segmentation network improved over feeding $I_f$ only. However, the segmentation network without $S_m$ served as a vanilla segmentation baseline, so it was included. We incorporated this observation to the revised Discussion.
>
> Regarding the literature of MTL, since the main focus of the paper is the joint registration segmentation application, we focused our literature on similar papers.
>
> Regarding the inference time, for non deep learning approaches it was in order of minutes, while for deep learning methods it was less than a second. This was also added to the revision.

---

> > ### Comment · AnonReviewer1 · 2020-04-03
> > **Thank you for the detailed answers to my comments.**
> >
> > 1. [Regarding the performance of bladder segmentation in Table 2...We added this observation to the Discussion.] - Thank you for adding this to the manuscript. This helps with the interpretation of the results.
> >
> > 2. [If you mean by sequential training, initializing the network with pre-trained task specific weights, we think that might bias the network towards a certain task over the other..] In Misra et al., they trained each network on each task then optimised the cross-stitch modules to decide how to share the learned representations. Table 3 in Misra showed that using pre-trained networks on the tasks outperformed for instance networks pre-trained on ImageNet. I think it would be interesting, at least in the future, to investigate if the learned representations differ between your training scheme and the original Cross-Stitch training.
> >
> > Thank you for your other additions to the Discussion.

---

### Official Review · AnonReviewer3 · 2020-03-14
**Borrowing an interesting idea from from multi-task learning in CV**

**Rating:** 3
**Confidence:** 4
**Recommendation:** Poster

**Summary:**

This paper investigates the joint learning of segmentation and registration tasks in the context of image-guided radiotherapy using daily prostate CT images. While for structures whose shape remains relatively close to the planning scan, and in regions of low contrast, registration performs best, segmentation can better adapt to changes in anatomy between visits (e.g. the bladder). This motivates the authors to leverage the strengths of both tasks. Therefore, it is proposed to employ a strategy that was originally proposed for multi-task learning in computer vision named cross-stitch. Cross-stitch layers are introduced to allow each of the two network paths (registration and segmentation), to leverage the features from the respective other task via a learnable weighting. The use of cross-stitch for segmentation of prostate CT scans has been compared to non-jointly learned task models (CNN based segmentation or registration), classic iterative image registration (elastix), and a hybrid method consisting of learned segmentation followed by classic iterative registration (elastix).

**Strengths:**

This paper is well written and the authors motivate the approach in the context of their application. While cross-stitch has been extensively evaluated on multiple computer vision tasks (dual task learning for two different related tasks), this paper introduces this idea to the medical community to jointly learn segmentation and registration, two pillars of medical image analysis, by intertwining both task-specific networks at the architectural level (in contrast to loss term only).

**Weaknesses:**

As the results on the independent test set demonstrate, and as openly discussed by the authors, the approach suffers from weak generalization to different scanner and/or institution (contrast change). This is a weakness of all compared learned approaches, except for the hybrid method of learned segmentation followed by classic iterative registration.

A weakness of the methodological contribution is the limited novelty, considering that this is merely the application of an idea proposed for multi-task learning in computer vision to the medical domain.

**Justification Of Rating:**

I see a main interest in this work for introducing the medical community to the idea of cross-stitch for multi-task learning, namely for joint registration and segmentation. The methodologically novelty is incremental, but combined with the comparison to alternative approaches, of sufficient interest to conference attendees.

**Paper Type:**

both

**Questions To Address In The Rebuttal:**

The authors claim in section 2.4. that the idea of hard parameter sharing for both tasks, except for the final output prediction layers, is a “novel joint network”. I would argue that this extreme case of multi-task learning has already been investigated in the original cross-stitch paper (cf. Misra et al., 2016, Fig 2a). Please remove this claim of "novelty".

Please specify what the chosen patch size n was in the experiments, which may explain the small batch size? Please comment on these choices.

**Special Issue:**

no

---

> ### Author Response · Authors · 2020-03-27
> **Answer to Reviewer 3**
>
> We would like to thank the reviewer for the positive feedback on paper clarity and the application.
>
> Regarding the performance of the learnable methods across the test dataset, we explained it in our reply to Reviewer 1 that the training datasets had a bladder filling protocol while the test dataset did not. This change in protocol corresponds to a big difference in the range of deformations, and therefore the network does not behave the same. We may address this issue by including synthetic large deformations during training.
>
> Regarding the novelty statement, we removed it in the revised manuscript. However, we believe it is novel in the context of the application.
>
> Regarding the patch size and number of batches, we chose a patch size of 96 $\times$ 96 $\times$ 96 and a batch size of effectively 4 (two unique training samples that are doubled by treating the fixed patches as moving patches and vice versa). We chose 96$^3$ since that would still produce a patch size of 17$^3$ at the lowest resolution which is reasonable to encode the deformation from the surrounding regions. We added this information to the revised version.

---

> > ### Comment · AnonReviewer3 · 2020-04-02
> > **Thanks for clarifications and amendments; maintaining rating**
> >
> > I would like to thank the authors for their reply to my questions. I maintain that this paper demonstrates an interesting application of cross-stitch units, originally introduced and investigated for multi-task learning of computer vision tasks, for joint registration and segmentation in medical imaging applications such as radiotherapy. Upholding the original rating and justification thereof.

---

### Official Review · AnonReviewer2 · 2020-03-17
**A novel approach combining cross-stitch network for segmentation and registration**

**Rating:** 3
**Confidence:** 5
**Recommendation:** Poster

**Summary:**

The paper presents a novel approach for simultaneous registration and segmentation of weekly CT scans for extracting organ at risk structures for prostate cancer. The method is written very well with reasonable experiments and results. Accurate segmentation of multiple OAR structures is essential for adaptive radiotherapy and this work if validated has application to several problems in radiation therapy.

A few things that could be improved and explained better are:
1. What was the reason to use a NCC metric. Its slow to compute and can be problematic when the images are far away?
2. Maybe I missed this,

**Strengths:**

A new approach that combines segmentation and registration. Use of cross-stitch network for this problem is novel. The paper is written reasonably well. The results seem to be improve over previous results.

**Weaknesses:**

The number of cases used for testing are few. While this method is supposed to present an approach for longitudinal registration and segmentation, results for longitudinal registration/segmentation as the organs deform and change over time is not shown.

Also, comparison methods don't use deep learning registration only methods like Quicksilver - Yang et.al.

Some details of the methods and reason for choice of architecture, ablation tests are missing. Ex. what is the patch size used? Why use leaky ReLU? What is the rationale. Authors say, they needed 4 cross-stitch units. The original Misra paper showed the importance of properly initializing the individual networks, studied the impact of learning rates as well as the initialization of the alpha values for cross-stitch. None of these seemed to have been studied here or at least were not reported. The only thing the authors talk about is that the number of cross-stitch units mattered the most.

Also, the authors talk about deep supervision but do not report on the effect of deep supervision or why it was needed. Ablation studies to parse out the impact of the various losses are needed.
Similarly, why NCC? Why not a better metric like EMD to compare patches? NCC is also slow and not necessarily accurate as EMD especially when the distributions are far away.
Finally, the planning CTs are quite a bit different from weekly CTs as the latter are often low-dose CTs used just for positioning. How well does NCC work here?


**Detailed Comments:**

Please see suggestions and comments above.

**Justification Of Rating:**

The paper presents a new approach for simultaneous registration and segmentation. This method is also applied to a significant clinical problem of weekly normal organ segmentation applied to radiotherapy. The methods are described reasonably well, albeit additional details should be included to make the paper more understandable. Furthermore, the results look convincing and clearly show improvements over some of the previous methods. However, deep learning registration only methods comparison would be more reasonable than Elastix registration. Ablation experiments are also needed for more comprehensive reporting of results. But this is not a journal paper, so some leeway in this regard could be given as long as rationale for the network design is presented.

**Paper Type:**

methodological development

**Questions To Address In The Rebuttal:**

Details of network initialization, at least some ablation results to show why 4 cross-stitch units are needed, rationale for NCC should be presented.
Also please at least explain why elastix and not a DL method was used for comparison.

**Special Issue:**

no

---

> ### Author Response · Authors · 2020-03-27
> **Answer to Reviewer 2**
>
> We would like to thank the reviewer for the positive feedback on paper novelty and clarity.
>
> Regarding our choice for using NCC loss instead of EMD, since the images are already affinely registered, NCC becomes a more reasonable, effective choice, that can be implemented efficiently. Note that in the (deep learning) registration literature this loss is abundantly used, much more than EMD.
>
> Regarding the concern of using low dose CT on the daily scans and how it could affect the performance, in the two datasets reported in the paper the planning and daily scans were acquired using full dose scans. However, in case of different distributions, EMD or mutual information metric could be better.
>
> Regarding the placement of cross-stitch units, we performed experiments by placing different numbers of cross-stitch units across the network (for example 14 or 8), but we found the performance was the best with 4. However, due to the limited paper space we could not report all the experiments. We place the 4 cross-stitch units after the 2 downsampling and the 2 upsampling layers. This is consistent with the findings of the original cross-stitch paper, where the authors experimented with placing cross-stitch units after every convolution activation map and after every pooling activation map, and found the latter performed better. We also updated the paper to emphasize that conclusion.
>
> Regarding the rationale for NCC loss, since the registration is done across a single modality with a similar intensity distribution, NCC is an obvious choice abundantly used in the registration literature. Moreover, the implementation is straightforward and efficient when using plain convolution operations.
>
> Regarding the question why we used elastix and not DL based algorithms for comparison; in the paper, we actually compared our algorithm against different algorithms from various categories, including DL: non-learning (elastix as a popular conventional tool); hybrid (Elmahdy et. al., 2019a), and deep learning-based (Elmahdy et. al., 2019b). In the ablation study also several DL methods are used.

---

> > ### Comment · AnonReviewer2 · 2020-03-30
> > **Author responses were satisfactory**
> >
> > Thank you for the clarifications for my questions. I think the paper is acceptable so long as the authors can add some of the clarification details regarding the architecture choice and cross-stitch unit explanations presented here.

---

### Author Response · Authors · 2020-03-27
**Comment from the authors**

We thank the reviewers for their insightful feedback. We are glad that the paper was well received and the novelty of the the application and the subsequent experiments was appreciated. The reviewers had some questions that we address in detail below.

---

### Meta-Review · Area_Chair1 · 2020-04-06
**MetaReview of Paper197 by AreaChair1**

**Rating:** 3
**Recommendation For Accepted Papers:** Poster

**Metareview:**

All the reviewers recommended acceptance of this work. After reading their comments and discussion with the authors, I think this work should be accepted for publication at MIDL.

Please, when submitting the Camera Ready version, take into account the suggestions made by the reviewers.

**Paper Type:**

both

**Special Issue:**

no

---

### Decision · Program_Chairs · 2020-04-11

Accept